# Latest Trends in Pollutant Accumulations at Threatening Levels in Energy-Efficient Residential Buildings with and without Mechanical Ventilation: A Review

**DOI:** 10.3390/ijerph19063538

**Published:** 2022-03-16

**Authors:** Hélène Niculita-Hirzel

**Affiliations:** Department of Occupational Health and Environment, Center for Primary Care and Public Health, Unisanté, University of Lausanne, CH-1066 Epalinges, Switzerland; helene.hirzel@unisante.ch; Tel.: +41-21-314-74-21

**Keywords:** indoor air quality, natural ventilation, mechanical ventilation, human, health outcomes, review

## Abstract

Improving the energy efficiency of buildings is a major target in developed countries toward decreasing their energy consumption and CO_2_ emissions. To meet this target, a large number of countries have established energy codes that require buildings to be airtight. While such a retrofitting approach has improved health outcomes in areas with heavy traffic, it has worsened the health outcomes in Nordic countries and increased the risk of lung cancer in areas with high levels of radon emissions. This review highlights the importance of adapting the characteristics of energy-efficient residential buildings to the location, age, and health of inhabitants to guarantee healthy indoor pollutant levels. The implementation of mechanical ventilation in new energy-efficient buildings has solved some of these problems; however, for others, a decrease in the level of outdoor pollutants was still required in order to achieve a good indoor air quality. A good balance between the air exchange rate and the air humidity level (adapted to the location) is key to ensuring that exposure to the various pollutants that accumulate inside energy-efficient buildings is low enough to avoid affecting inhabitants′ health. Evidence of the protective effect of mechanical ventilation should be sought in dwellings where natural ventilation allows pollutants to accumulate to threatening levels. More studies should be carried out in African and Asian countries, which, due to their rapid urbanization, use massive volumes of unproven/unrated building materials for fast-track construction, which are frequent sources of formaldehyde and VOC emissions.

## 1. Introduction

To limit global warming, the building and construction sector must drastically reduce its carbon emissions. This sector was identified as accounting for 36% of final energy use and 39% of energy- and process-related emissions in 2017 [1]. From the 1970s, the energy-efficient renovation of existing buildings has been encouraged by regulatory and governmental incentives in developed countries [2,3]. In addition, to meet energy and climate-change targets, energy-efficiency requirements were introduced in European building codes for all new buildings [3,4,5]. While building policies support improving both the insulation and airtightness of windows and doors, they differ with regard to incorporating a ventilation concept, which is not required in retrofitted buildings, but is a requirement for new buildings. By not integrating a ventilation concept, the airtightness of retrofitted buildings results in a reduction in the ventilation rate (to approximately 4 L/s per person or less [6]), which has dramatic consequences for the indoor air quality (IAQ) and moisture load. However, the inadequate functioning of mechanical ventilation in new energy-efficient buildings can also have similar negative consequences for the IAQ. The lack of an adequate ventilation was identified in a recent exhaustive review to be the most influential risk factor for respiratory disease in a household environment [7]. The need for healthy indoor air by adapting the ventilation was supported not only by researchers, but also by professionals, especially since the beginning of COVID-19 pandemic. The existence of numerous guidelines established for air quality and ventilation (ASHRAE standards 62.1 and 62.2 [8]) helped practitioners to widely adopt them in this particular context. Moreover, the COVID-19 pandemic increased the number of professionals who are aware of the importance of these building parameters and their effect on occupant health in comparison to other attributes [9]. Such an opportunity must be taken to support the professionals in improving the indoor air quality in dwellings. To assist in decision making, the parameters proposed by Asim et al. will be useful in guiding professionals to select the optimum heating, ventilation, and air conditioning (HVAC) system [10]. Despite this, the residential housing stock is likely to be considered less in retrofitting strategies, mostly due to the lack of economic resources of the owners or the difficulty of moving people from rented houses during the retrofitting process. Therefore, it is essential to continue to assess the IAQ in residential buildings in order to raise awareness in decision making for this type of building as well. Air pollution can be defined as the presence of substances in the air that are harmful to humans. A large number of pollutants—vapor or particle matter (PM)—emitted indoor by building materials and human activities, or penetrating from outdoors—have been found to accumulate in residential indoor air. The major sources of the most common pollutants observed in indoor air were recently reviewed by Tran et al. [11]. Those pollutants can be deeply inhaled, and their short- and long-term effects on health have been well documented [12]. Thus, long-term exposure to elevated concentrations of radon or particulate matter (PM)—coarse (PM_10_) and fine (PM_2.5_)—has been associated with an increased risk of premature death from lung cancer or cardiovascular disease, respectively. Short-term exposure to PM_2.5_ has been associated with an increase in the frequency of asthma attacks, particularly in children and the elderly. Additionally, mold growing on built surfaces has been associated with the development of asthma in young children and the exacerbation of asthma or chronic coughing in adults. Short-term exposure to other indoor pollutants, such as volatile organic compounds (VOCs), may also cause eye, nose, and throat irritation; shortness of breath; headaches; fatigue; nausea; dizziness; and skin problems. Altogether, exposure to air pollution is estimated to cause millions of deaths per year and the loss of healthy years of life. The burden of disease attributable to air pollution was recently estimated to be on a par with unhealthy diets or tobacco smoking. Examples of the diversity of health problems that are induced by exposure to indoor pollutants are shown in Table 1.

Based on the epidemiological evidence, air quality guidelines were established in 1987 by the World Health Organization (WHO) and different relevant organizations, [11] and subsequently endorsed by at least 194 countries [13]. However, those guidelines were established only for buildings such as residential houses, schools, hospitals, public buildings, and offices, and they were adapted by each country to their own particular circumstances. The lowest air-quality guidance levels recommended for the common indoor air pollutants are indicated in Table 1.

To control the risk of exposure to residential indoor air pollutants, the outdoor and indoor sources of these pollutants have been intensively researched and successfully identified [11,13]. Therefore, solutions have been proposed to eliminate certain sources (e.g., naphthalene and formaldehyde) or to limit the infiltration or indoor accumulation of those that cannot be eliminated by adapting the building characteristics. Thus, the airtightness of energy-efficient buildings—whether retrofitted or new—is expected to limit the infiltration of outdoor pollutants, whereas the combination of this characteristic with efficient mechanical ventilation is expected to prevent the accumulation of indoor pollutants. However, the heterogeneity of existing studies that explored the relationships between IAQ and energy-efficient dwellings with natural or mechanical ventilation, as recently pointed out by Moreno-Rangel et al. [16], suggests a more complex scenario. While some suggested that energy-efficient dwellings characteristics may improve IAQ, others have found overheating and high levels of CO_2_. The cause of this disparity was suggested to be due to deficiencies in the ventilation systems. Nevertheless, it remains to clarify to what extent such a heterogeneity exists among studies that considered residential buildings with similar energy-efficient standards.

The aim of this review was to:− Focus on the levels of pollutants present in the indoor air of residential dwellings built or retrofitted under similar energy-efficient codes;− Identify the differences in the levels of those pollutants between buildings stocks with distinct ventilation systems;− Analyze if these differences were or were expected to be associated with a difference in health outcomes.

Section 3.1 reviews the existing data on outdoor pollutants infiltration, while Section 3.2 presents a synthesis of those on indoor pollutants associated with health outcomes.

## 2. Material and Methods

### Search Strategy

An electronic search was conducted using Web of Science and PubMed Central for relevant articles published in peer-reviewed journals by combining the following keywords: “energy efficient” OR energy-efficient, residential OR homes OR home OR dwellings, ventilation, health OR “health outcomes”, “indoor air quality” OR radon OR PM_10_ OR PM_2.5_ OR ozone OR nitrogen dioxide OR VOCs OR formaldehyde OR trichloroethylene OR indoor moisture OR microbial growth OR moulds OR molds. The references obtained were screened for the presence of the main keywords in the titles and abstracts. Inclusion criteria included were: (a) written in English; (b) on residential dwellings that were new and retrofitted with an energy-efficient standard; (c) a single-family home; and (d) with at least one indoor air pollutant measured. Exclusion criteria included: (a) buildings for other usage than residential (offices, schools, hospitals); (b) only health outcomes reported; and (c) modeling of pollutant distribution. We retained the articles that studied houses built or retrofitted after 2010 according to very-low-energy or passive house standards.

## 3. Results and Discussion

### 3.1. Pollutants Studied in Residential Energy-Efficient Buildings after 2010

The literature review revealed 24 articles that quantified pollutants in the indoor air of residential energy-efficient buildings. PM_2.5_ was the most frequently monitored pollutant, with a wide geographical distribution—four studies were run in the USA, two in South Korea, four in different European countries (France, Italy, Finland, and Lithuania). Radon was monitored in eight studies—seven in different European countries and one in the USA. The VOCs were quantified in eight studies—five in European countries (France, Austria, and Switzerland), two in the USA, and one in South Korea. Formaldehyde was monitored in the largest number of countries: the USA (3 studies), the United Arab Emirates (1 study), South Korea (1 study), China (1 study), Austria (1 study), and France (1 study). NO_2_ levels in indoor air were monitored in only four studies—two in the USA and two European (one in a French building stock, and one in Finnish and Lithuanian stocks). Ozone was quantified in indoor air in only one study that compared a Finnish and a Lithuanian stock of dwellings. Concentrations of fungal spores were reported in five studies, all European (Austria, France, and Switzerland). The relative humidity of the air was also studied in two Asiatic countries (Bahrain and South Korea).

### 3.2. Efficiency of Energy-Efficient Buildings to Limit Outdoor Pollutant Infiltration

In indoor environments, concentrations of pollutants originating from outdoor air are influenced by their outdoor spatiotemporal patterns of concentration and by the proximity of the building to outdoor sources (e.g., roads with heavy traffic). While indoor pollution concentrations depend on the amount of air pollution penetrating from outdoors, they also depend on the efficiency of the ventilation and, for gaseous air pollutants, on the reaction rates with indoor surfaces. Thus, air pollutants with a lower penetration factor such as PM_2.5_ remain suspended indoors longer than gases such as O_3_, which has a high reaction rate with indoor surfaces [17]. While the role of these different factors has been compared between conventional, retrofitted, and new energy-efficient buildings in different countries on different continents [18,19,20,21,22], the health issues related to modulating the concentration of some of these pollutants has rarely been addressed.

#### 3.2.1. Efficiency of Energy-Efficient Buildings to Limit Radon Accumulation Indoors

Radon is a radioactive gas naturally emitted by soils at variable concentrations depending upon the topography and soil structure. For example, Germany, Poland, the Netherlands, most French regions, and the UK have lower levels of radon emissions, on average, than Austria, Finland, Sweden, Switzerland, or the Czech Republic. This pollutant infiltrates indoors and can accumulate to life-threatening concentrations when the ventilation is poor. Due to the carcinogenic effects of radon, its concentration in indoor air is recommended to be lower than 100 Bq/m^3^ if possible [13]. Nevertheless, the action level for reduction is frequently set between 200 Bq/m^3^ and 400 Bq/m^3^. At the same time, all new buildings are expected to have concentrations below 100 Bq/m^3^. These expectations were confirmed in a French stock of dwellings [18] and an Austrian stock [20] for energy-efficient dwellings built after 2010. The radon level was within the same range in both studies (17–31 Bq/m^3^ and 24–29 Bq/m^3^, respectively), and rarely exceeded the WHO reference level of 100 Bq/m^3^ [13,15]. In areas that have a low risk of radon emissions, the modern characteristics of dwellings built after 2010—even without mechanical ventilation—seem sufficient to avoid radon accumulate to life-threatening concentrations. In contrast, in areas that have a high risk of radon emissions, the implementation of mechanical ventilation with heat recovery had a better performance in radon reduction. Relative to naturally ventilated residences, Austrian and Swiss dwellings with mechanical ventilation systems have been found to present a significantly lower radon concentration [20,23]. This was particularly true for dwellings located in areas with a very high risk of radon emission, in which mechanical ventilation increased the difference in radon concentrations between mechanically and naturally ventilated houses (geo-mean: 96 vs. 251 Bq/m^3^, *p* < 0.001) [23]. Nevertheless, the Swiss dwellings accumulate slightly more radon than Austrian ones, in both mechanically and naturally ventilated dwellings (geo-mean: 58 and 105 Bq/m^3^, respectively) [23]. At the same time, thermal retrofitting without the implementation of mechanical ventilation in the houses located in areas with such a high risk must be avoided. Airtightness of buildings has been negatively linked to indoor radon levels [23]. However, while centralized and decentralized mechanical supply and exhaust ventilation with heat recovery yielded a good efficiency in radon reduction, the best performance was confirmed to be based on subslab depressurization (SSD) [24]. The inconvenience of implementing SSD systems in inhabited houses is related to the cost and the disruption of daily household activities. These findings highlight the importance of adapting a retrofitting strategy to local radon emission levels in order to protect residents′ health.

#### 3.2.2. Efficiency of Energy-Efficient Buildings to Limit PM_2.5_ Infiltration

Another pollutant with recognized short- and long-term impacts on human health is particulate matter (PM) with an aerodynamic diameter smaller than 10 microns. PM emissions are associated with windblown desert dust and anthropogenic activities such as road traffic, but also with cigarette smoke, and solid fuel or wood heating (reviewed in [12]). While both PM_10_ and PM_2.5_ were recognized to impact human health, data in energy-efficient residential dwellings were reported only on PM_2.5._ Ambient PM_2.5_ concentrations are known to vary substantially between and within regions of the world, and evolve with time. While they decreased in the WHO European Region, the WHO Region of the Americas, and the WHO Western Pacific Region in the recent years, they increased elsewhere in the world. This was partially due to a difference in the PM_2.5_ sources. Therefore, to control the level of this pollutant in indoor air, it is necessary to adapt an intervention in response to the PM_2.5_ sources. In dwellings located in urban areas with heavy traffic, the infiltration of PM_2.5_ from outdoors must be reduced [25,26]. In dwellings where people are using solid fuel, wood heating, or continuing to smoke inside, the users’ habits must be changed. Indeed, a high air exchange rate in areas of heavy traffic was associated with adverse respiratory outcomes [26,27]. The airtightness of the last generation of energy-efficient dwellings (European standard) is an efficient tool for limiting PM_2.5_ infiltration indoors [28]. Even more so, the implementation of mechanical ventilation reduced the PM_2.5_ accumulation indoors in both recent and retrofitted energy-efficient dwellings, in particular during the heating period [18,27,28]. Nevertheless, the difference in the PM_2.5_ level between mechanically and naturally ventilated energy-efficient dwellings is generally small. The PM_2.5_ median indoor levels was below the WHO guideline (5 μg/m^3^) only in a Finnish stock of dwellings (4.3 μg/m^3^), while it was below 10 μg/m^3^ of annual exposure in a Lithuanian stock of dwellings [27] and in an American one [29], but was above this target in French or Italian stock with similar characteristics (13 μg/m^3^ and 15 µg m^−3^, respectively, during the winter) [18,30]. Interestingly, the mechanical ventilation was reported to decrease PM_2.5_ median indoor levels to 7.5 in comparison to 13.4. The health issues related to modulating the concentration of outdoor PM_2.5_ in indoor air was addressed in a Korean energy-efficient stock of buildings with similar energetic standards [28]. The PM_2.5_ level was lowered enough by the mechanical ventilation versus the natural ventilation (6.0 ± 6.9 μg/m^3^ vs. 8.7 ± 8.6 μg/m^3^, respectively [28]) to observe a decrease in allergic rhinitis incidence in adults. However, this decrease was insufficient to prevent the incidence of allergic rhinitis and atopic dermatitis in children. Complementary measures must be taken to keep the PM_2.5_ low enough to avoid detrimental effects on health. One approach is to decrease outdoor pollution by increasing the density of energy-efficient buildings [31], while another involves directly substituting for coal in its use for power in industry and households [32].

#### 3.2.3. Efficiency of Energy-Efficient Buildings to Limit Gaseous Outdoor Pollutant Accumulations

Nitrogen dioxide (NO_2_) is another pollutant with a well-established guideline value. The patterns of its ambient concentrations are quite different from those of PM_2.5_ and ozone (O_3_). The highest NO_2_ exposure levels were observed in eastern Asia, the Middle East, North America, and much of Europe. Nitrogen dioxide also displays a distinct urban–rural gradient, with higher concentrations in more densely populated urban areas [18]. This pattern contrasts distinctly from that of PM_2.5_, which is more homogeneous regionally due to its longer atmospheric lifetime and diversity of (urban, rural, and regional) sources, and ozone, which displays higher concentrations downwind of urban areas. Up to 50% of ambient NO_2_ was estimated to infiltrate in dwellings [17]. However, in energy-efficient houses, mechanical ventilation, as well as the changes in energy-efficient building codes, did not seem to influence the NO_2_ indoor level [18,27,29]. The interest in considering the NO_2_ quantification in further studies is based on its coupling relationship with other pollutants such as ozone (O_3_). Ozone, in presence of NO_2_, promotes SO_2_ oxidation, resulting in the formation of particulate sulfate. Furthermore, O_3_ concentration affects the photochemical reaction process of NO_2_. In spite of this important reaction, very few studies quantified O_3_ concentration in the indoor environment of energy-efficient dwellings with a residential usage. Thus, we identified only one epidemiological study that looked for an association between O_3_ concentrations and the ventilation rate in energy-efficient dwellings [27]. There is a definite need to consider this pollutant in further epidemiological studies to estimate whether the characteristics of the most recent generation of energy-efficient buildings are sufficient to keep the levels of this pollutant low enough in the indoor environment to avoid affecting inhabitants’ health.

### 3.3. Efficiency of Energy-Efficient Buildings to Limit Indoor Pollutant Accumulations

While the airtightness of retrofitted energy-efficient buildings might limit the infiltration of outdoor pollutants, the resulting reduced ventilation can have unwanted repercussions on indoor air quality (IAQ) [33] and adverse effects on respiratory health [34,35,36]. When the household air changes per hour (ACH) fall below the European standard of 0.5 ACH [37], this results in the accumulation of most indoor pollutants—including bacteria that are closely related to human pathogens [38]—and an increase in the relative humidity of the ambient air, which favors the development of molds [36,39]. Indoor air temperature also matters, as it regulates the relative humidity content and can promote the release of pollutants from building materials.

#### 3.3.1. Efficiency of Energy-Efficient Buildings to Exhaust Humidity

Exposure to molds was associated with increased incidence of asthma and allergic symptoms [40,41,42]. In addition to its negative effects on health, dampness affects the durability of materials and favors interstitial condensation. Taken together, all of these negative effects have prompted the implementation of ventilation requirements—whether natural or mechanical—in building energy standards [43]. The more efficient the ventilation system, the larger the decrease in the relative humidity of the ambient air. However, the implementation of mechanical ventilation does not guarantee an efficient air exchange rate, not only due to technical deficiencies (such as unbalanced ventilation systems), but also due to human interaction with the systems (reviewed in [44]). In parallel, natural ventilation has been shown to offer sufficient air exchange in green buildings (that were conceived with this orientation in mind) in areas with mild summers [45]. Therefore, small or nonexistent differences in the air relative humidity (varying between 45% and 50%) might be observed between mechanically and naturally ventilated energy-efficient dwellings [20,28]. However, even a slight decrease in the air humidity level in mechanically vs. naturally ventilated dwellings may be sufficient to prevent the development of certain fungal species on building surfaces, such as the species indicative of water damage, *Ulocladium* and *Stachybotrys*, or pathogenic species of the *Aspergillus* complex [46]. In addition, mechanical ventilation is also efficient in lowering the concentration of airborne fungal spores [20,21,46,47]. Consequently, any increase in the ventilation rate is expected to reduce both the prevalence of respiratory symptoms (e.g., asthma, allergic rhinitis, and sinusitis) and the incidence of severe respiratory infections that are caused by *Aspergillus* (reviewed in [48]). Nevertheless, to have a healthy house, no mold development should occur during the construction process. Such an event is favored by the lack of ventilation and the airtightness of the latest generation of energy-efficient buildings [49]. If this occurs, it could persist during the whole life of building. When molds develop on surfaces, extensive environmental remediation and replacement of the infected materials is recommended for their removal. The use of a chemical treatment reduces exposure to molds and their byproducts (the mycotoxins), although it does not eradicate them [50]. The effectiveness of each remediation approach in improving health outcomes must be supported by epidemiological studies, and to the author’s best knowledge, almost all data on the effectiveness of extensive remediation methods were evaluated here [51]. However, it must be ensured that the air relative humidity is not excessively lowered, as skin dryness and eye fatigue were regularly reported in such cases [28,52].

#### 3.3.2. Efficiency of Energy-Efficient Buildings to Limit VOCs Accumulation

Mechanical ventilation of buildings not only results in decreased concentrations of fungal and bacterial colony-forming units, but also decreases in the indoor levels of other pollutants, such as volatile organic compounds (VOCs), in particular formaldehyde. However, while this decrease has been systematically reported in recent years [20,30,53,54], very few studies examined the associations between respiratory health and exposure to VOC accumulations in residential indoor air, in particular in energy-efficient dwellings [28,52,54].

Formaldehyde was classified as a Group 1 human carcinogen by the International Agency for Research on Cancer in 2004. To protect against both acute and chronic sensory and airway irritation in the general population, the WHO issued an air-quality formaldehyde guideline threshold of 100 μg/m^3^. In an indoor environment, the emission rate of formaldehyde greatly depends on the building and furniture materials. Nevertheless, an efficient ventilation can limit its accumulation. We found that the formaldehyde levels in residential energy-efficient-dwellings greatly differed between the established cities and regions with a rapid urbanization process. Thus, the formaldehyde levels reported in the USA, Canada, and different European countries were generally lower than the WHO AGL (100 μg/m^3^) or even the Canadian AGL (50 µg/m^3^) [29,52,54], while those reported in Asian countries were generally higher than the AGL [55,56]. Nevertheless, none of the epidemiological studies conducted on energy-efficient dwellings were done in these at-risk populations. The few existing ones were conducted in dwellings with low levels of formaldehyde. These studies showed that, while formaldehyde and, more generally, total VOC concentrations, were lower in dwellings with mechanical ventilation systems than in those with only window ventilation, both remained generally in the range of the air-quality guidance levels. Therefore, when a weak but statistically significant correlation between the frequency of symptoms (dizziness, nausea, and headaches in adults [52]; emotional distress in the elderly [54]) and the concentration of formaldehyde was reported, this correlation was independent of the type of ventilation systems [52]. Therefore, health effects will benefit from being monitored in Asian or African countries with a rapid urbanization process, where very high levels of formaldehyde have been reported indoors. The only existing Asian epidemiological study, conducted in South Korea, did not specifically monitor formaldehyde, but total VOC. In this study, total VOC concentrations in dwellings with mechanical ventilation systems were lower than in the properties with only window ventilation [28]. Nevertheless, they both remained generally in the range of the air-quality guidance levels (total VOC: 0.2–0.6 mg/m^3^ (FISIAQ)). The authors detected a significant contribution of the level of exposure to total VOC in the development of an allergic rhinitis in adults, but not specifically associated with the presence of mechanical ventilation [28]. Consequently, evidence of the protective effect of mechanical ventilation should be sought in dwellings where natural ventilation allows pollutants to accumulate to threatening levels.

## 4. Conclusions

There is a lack of studies on indoor-related risk factors in energy-efficient residential buildings, partially due to an inhomogeneous distribution of energy-efficient dwellings over the world that is concentrated almost entirely in Europe and North America. The implementation of energy-efficient building standards for residential buildings in Singapore (2012), Vietnam (2013), the People’s Republic of China (2016), Mexico (2016), and India (2018) [57] is expected to favor an increase in the energy-efficient stock of residential dwellings in these countries. Therefore, studies on the indoor air quality in such buildings should be encouraged and supported. This need in African and Asian countries is justified by their rapid urbanization, which uses massive volumes of unproven/unrated building materials for fast-track construction—frequent sources of formaldehyde and VOC emissions [55]. On the other hand, epidemiological studies in European countries must consider correlating health outputs and exposure to “new” pollutants such as phthalate ester plasticizers, brominated flame retardants, nonionic surfactants, and coalescing solvents. New challenges in protecting the health of residents await us with the implementation of new building materials that minimize the carbon footprint of buildings, reduce vulnerability, and increase the resilience of buildings to climate change.

## Figures and Tables

**Table 1 ijerph-19-03538-t001:** Examples of pollutants with the lowest air quality guidelines for indoor air, sources, and health effects.

Pollutant	Outdoor Sources	Indoor Sources	Risk of Health Effects	Excess Lifetime Risk	Air-Quality Guidance Level
Radon	Decay of radium in the soil subjacent to a house	Concrete; sandstone; burned and unfired brick; marble; granite	Lung cancer	For 1/100 and 1/1000: 67 and 6.7 Bq/m^3 a^ for current smokers, respectively; and 1670 and 167 Bq/m^3^ for lifelong nonsmokers, respectively	100 Bq/m^3^ (2.7 pci/L ^b^) [13]
Particulate matter (PM_2.5_)	Combustion processes from motor vehicles; solid fuel burning; industry	Combustion sources for cooking and heating; tobacco smoke	Acute lower respiratory infections; chronic obstructive pulmonary disease; ischaemic heart disease; lung cancer; stroke		5 μg/m^3 c,^* [13] 15 μg/m^3^ ** [13]
Ozone (O_3_)	Photochemical reactions of O_2_, (NOx), and VOCs	Photocopying; air purifying; disinfecting devices	Respiratory symptoms; acute lower respiratory infections; aggravate lung diseases		40 μg/m^3^ [14] 100 μg/m^3^, 8 h ** [13]
Nitrogen dioxide (NO_2_)	Road traffic	Gas stoves; tobacco smoke	Causes respiratory effects (asthma exacerbation)		10 μg/m^3^ * 25 μg/m^3^ ** [13]
Sulfur dioxide (SO_2_)	Burning of high-sulfur coals; heating oils in power plants; industrial boilers; metal smelting		Irritation of the nose, eyes, throat, and lungs		40 μg/m^3^ ** [13]
Carbon monoxide (CO)	Heavy traffic; attached garages	Combustion sources for cooking and heating; tobacco smoke	Reduction in exercise tolerance; increase in symptoms of ischaemic heart disease		4 mg/m^3 d,^** [13]
Benzene	Heavy traffic; attached garages; petrol stations; certains industries	Building materials that off-gas benzene; furnishing materials; human activities; heating and cooking	Leukemia	For 1/10,000, 1/100,000, and 1/1,000,000: 17, 1.7, and 0.17 μg/m^3^, respectively	No safe level [13]
Formaldehyde	Fuel combustion from traffic	Building materials and products; furniture and wooden products containing formaldehyde-based resins; tobacco smoke	Sensory irritation of eyes; increases in eye-blink frequency; conjunctival redness		50 µg/m^3^ * (40 ppb) [14] 100 μg/m^3^ (81 ppb ^e^) for 30 min [13]
Naphthalene	Heavy traffic; petrol stations; oil refineries	Mothballs; unvented kerosene heaters; tobacco smoke	Respiratory tract lesions		10 μg/m^3 a^ [13,14]
Polycyclic aromatic hydrocarbons (e.g., benzo[a] pyrene)	Heavy traffic	Cooking and heating with solid fuels	Lung cancer	For 1/10,000, 1/100,000, and 1/1,000,000: 1.2, 0.12, and 0.012 ng/m^3^, respectively ^f^	No safe level [13]
Trichloro-ethylene		Water ingestion; dermal absorption when showering; breathing indoor air	Cancer (liver, kidney, bile duct, and non-Hodgkin′s lymphoma)	For 1/10,000, 1/100,000, and 1/1,000,000: 230, 23, and 2.3 μg/m^3^, respectively	No safe level [13]
Tetrachloro-ethylene	Dry-cleaning facilities		Early renal disease and impaired performance		0.25 mg/m^3 a^ [13]
Indoor moisture; microbial growth		Water damage; leakage; moisture	Development and exacerbation of respiratory diseases (e.g., asthma)		<500 CFU/m^3^ [15]

* Annual; ** 24 h, a few days per year; ^a^ becquerels per cubic meter of air; ^b^ pounds per square inch per liter of air; ^c^ micrograms per cubic meter of air; ^d^ milligrams per cubic meter of air; ^e^ parts per billion; ^f^ nanograms per cubic meter of air.

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
