# Peer review of "Latest Trends in Pollutant Accumulations at Threatening Levels in Energy-Efficient Residential Buildings with and without Mechanical Ventilation: A Review"

_ijerph, 2022, doi:10.3390/ijerph19063538_

Round 1

Reviewer 1 Report

The response to previous comments was not sufficient, each point should be addressed explicitly and any changes notes clearly for reviewers. 

The manuscript still needs significant improvement. The topic is important but the approach in this review article is not systematic enough.

It should be detailed how the review was conducted. There appear many missing references that may be relevant, and the manuscript generally just picks relevant citations. There appears to be no systematic approach to ensure the evidence presented is balanced and robust. 

Dwellings are not just simply mechanically or naturally ventilated. With this there are a range of systems, operation strategies and air tightness standards. There is a significant limitation in reviewing studies from diferent countries where the context is so different. 

The balance of indoor and outdoor sources of pollutants is not suffciently addressed. This depends on ventilation rates, filtration and the respective sources. The research question is presumably "what are the differences in pollutant accumulation and related health outcomes between mechanically and naturally ventilated dwellings". More needs to be done to actually frame evidence, and just as importantly gaps, against this. 

The conclusions are opinions, not backed up sufficiently by the preceeding review. In the conlcusions I would expect to see statements summarising the evidence for the research question of the paper. 

Author Response

The response to previous comments was not sufficient, each point should be addressed explicitly and any changes notes clearly for reviewers. The manuscript still needs significant improvement. The topic is important but the approach in this review article is not systematic enough.

Answer. I thank the reviewer to help us improve this manuscript by his/her comments. The methods of the systematic research of articles is indicated and the aim of the review was detailed. We are mentioning now that the aim of this review is to:

- focus on the levels of pollutants present in the indoor air of residential dwellings built or retrofitted under similar energy efficient codes,

- identify the difference in the level of those pollutants between buildings stocks with distinct ventilation systems,

- analyse if this difference is or is expected to be associated with a difference in health outcomes. 

It should be detailed how the review was conducted.

Answer: Really sorry to miss this part. A material and methods section was added to present how we conduct the systematic review. We also add a  paragraph in the Results section to present the results of the systematic review.

There appear many missing references that may be relevant, and the manuscript generally just picks relevant citations. There appears to be no systematic approach to ensure the evidence presented is balanced and robust. 

Answer: I hope that the clarification of the aim of the article and the presentation of articles review process, better explain the choice of the citations.

Dwellings are not just simply mechanically or naturally ventilated. With this there are a range of systems, operation strategies and air tightness standards. There is a significant limitation in reviewing studies from diferent countries where the context is so different. 

Answer: I completely agree with the reviewer. There is for this reason that we pick up studies conducted on residential buildings with similar energy-efficient standards. This explain the limited number of studies that were presented in the present review. We pay attention that this information is appearing now in the main text.

The balance of indoor and outdoor sources of pollutants is not suffciently addressed. This depends on ventilation rates, filtration and the respective sources. The research question is presumably "what are the differences in pollutant accumulation and related health outcomes between mechanically and naturally ventilated dwellings". More needs to be done to actually frame evidence, and just as importantly gaps, against this. 

Answer: I completely agree with the reviewer. However, those sources were largely reviewed elsewhere. There is for this raison that we didn’t develop this aspect in the present review. Nevertheless, we cited the reviews that they do it.

The conclusions are opinions, not backed up sufficiently by the preceeding review. In the conlcusions I would expect to see statements summarising the evidence for the research question of the paper. 

Answer: I completely agree with the reviewer. A new conclusion is proposed in the present version that I hope reach the reviewer expectations.

Reviewer 2 Report

Energy-efficient buildings with and without mechanical ventilation: differences in pollutant accumulation and related health outcomes

Page 1

 4 L/s is used. Please use full form first time, then use abbreviation.

Page 2

Bq/m3 Please use full form first time.

pci/L Please use full form first time.

µg/m3 Please use full form first time.

8h please use full form first time.

Page 3

81 ppb please use full form first time.

ng/m3 please use full form first time.

mg/m3 please use full form first time.

Author Response

Answer: I thank the reviewer to help us improve this manuscript by his/her comments.

Page 1

 4 L/s is used. Please use full form first time, then use abbreviation.

Answer: Done in the text

Page 2

Bq/m3 Please use full form first time.

Answer: As for the first time Bq/m3  appears in a table, the full form was added as a footnote of the table. I hope that the reviewer agree with this way of annotation

pci/L Please use full form first time.

Answer: As for the first time pci/L appears in a table, the full form was added as a footnote of the table. I hope that the reviewer agree with this way of annotation

µg/m3 Please use full form first time.

Answer: As for the first time µg q/m3  appears in a table, the full form was added as a footnote of the table. I hope that the reviewer agree with this way of annotation

8h please use full form first time.

Answer: Done in the table.

Page 3

81 ppb please use full form first time.

Answer: As for the first time ppb appears in a table, the full form was added as a footnote of the table. I hope that the reviewer agree with this way of annotation

ng/m3 please use full form first time.

Answer: As for the first time ng/m3  appears in a table, the full form was added as a footnote of the table. I hope that the reviewer agree with this way of annotation

mg/m3 please use full form first time.

Answer: As for the first time mg/m3  appears in a table, the full form was added as a footnote of the table. I hope that the reviewer agree with this way of annotation

Reviewer 3 Report

1. Please note that this study is timely, important and innovative. On the other hand, the references and contents of the Table 1 need improvement, such as concrete, sandstone, burned and unfired brick, marble, and granite are thought to be the significant source of indoor radon.
2. Indoor moisture has the limit value of 500 CFU/m3in hygienic specification for public building in the WHO’s guideline published in 1990 (World Health Organization, 1990). Are there any standard for indoor air quality? Please confirm this point.
3. This manuscript only focused on PM2.5, however, indoor PM10 was also as an important air pollutant, why did not focus on the impact of PM10?
4. In the last sentence of the second paragraph, “The WHO air quality guidance levels are usually the lowest among those recommended by different agencies”, is it appropriate to write this way, for all indicators? please explain it.
5. Current COVID-19 pandemic has put a spotlight on the transmission of infectious diseases brought by pathogenic airborne microbes, though this manuscript did not introduce in microbiology, and It would be brilliant if airborne microbiology added.

Author Response

1. Please note that this study is timely, important and innovative. On the other hand, the references and contents of the Table 1 need improvement, such as concrete, sandstone, burned and unfired brick, marble, and granite are thought to be the significant source of indoor radon.

Answer: I thank the reviewer for this remark. The sources of indoor radon were added

2. Indoor moisture has the limit value of 500 CFU/m3 in hygienic specification for public building in the WHO’s guideline published in 1990 (World Health Organization, 1990). Are there any standard for indoor air quality? Please confirm this point.

Answer: I agree that 500 CFU/m3 guideline is given by WHO, Singapore (1996). However Canada Public Works (2005) suggests that Indoor fugal quantities should be lower compared to outside, while ACGIH (1989) recommended 200 CFU/m3 as a guideline for fungal bioaerosols. I agree that these are not guidelines from official agencies. Therefore the 500 CFU/m3 guideline was added in the table.

3. This manuscript only focused on PM2.5, however, indoor PM10 was also as an important air pollutant, why did not focus on the impact of PM10?

Answer: I completely agree with the reviewer. However, astonishing none of the studies on energy-efficient residential dwellings built after 2010 considered to monitor the PM10 in indoor air. This is mention now in the main manuscript.

4. In the last sentence of the second paragraph, “The WHO air quality guidance levels are usually the lowest among those recommended by different agencies”, is it appropriate to write this way, for all indicators? please explain it.

Answer: I agree that the wording is awkward. The sentence was reformulated. I agree that WHO don’t recommend systematically the lowest guidance levels among those recommended by different agencies

5. Current COVID-19 pandemic has put a spotlight on the transmission of infectious diseases brought by pathogenic airborne microbes, though this manuscript did not introduce in microbiology, and It would be brilliant if airborne microbiology added.

Answer: I fully agree with this aspect. However, all studies on the role of ventilation on the concentration and spread of viral propagules in indoor air have been carried out in offices, schools, public buildings, but not in residential dwellings – the type of dwellings to which we limit our review. Nevertheless, a paragraph on the role of the pandemic in raising professional awareness on the issue of air quality has been added in the introduction section.

Reviewer 4 Report

the paper is a qualitative interesting evaluation of interaction between mechanical ventilation, health issues related to air quality, and energy efficiency. while the paper deals with an interesting topic, it does not present any original contribution related to these concepts, since it assesses the variety of results without providing any new data. to this aim, the author is recommended to write a lit review paper and to cancel this submission as original article. to the aim of the research literature review, the author should better evaluate all the existing literature about technical data on both the engineering and life science perspective.

Author Response

the paper is a qualitative interesting evaluation of interaction between mechanical ventilation, health issues related to air quality, and energy efficiency. while the paper deals with an interesting topic, it does not present any original contribution related to these concepts, since it assesses the variety of results without providing any new data. to this aim, the author is recommended to write a lit review paper and to cancel this submission as original article. to the aim of the research literature review, the author should better evaluate all the existing literature about technical data on both the engineering and life science perspective.

Answer: I thank the reviewer to help us improve this manuscript by his/her comments.

The manuscript was formatted as a review. The aim of the review was clarify. We are mentioning now that the aim of this review is to:

- focus on the levels of pollutants present in the indoor air of residential dwellings built or retrofitted under similar energy efficient codes,

- identify the difference in the level of those pollutants between buildings stocks with distinct ventilation systems,

- analyse if this difference is or is expected to be associated with a difference in health outcomes. 

The articles selection was detailed in the material and methods to justify the paper selection.

Recent reviews focusing on the technical data on both the engineering and life science perspective were cited.

This manuscript is a resubmission of an earlier submission. The following is a list of the peer review reports and author responses from that submission.

Round 1

Reviewer 1 Report

Thank you for giving me the opportunity to review this well-written and concise paper about the current state of knowledge in this very relevant field; it should be of interest to a wide interdisciplinary audience. For completeness´sake it would not hurt to expand briefly on the relation to climate change and sustainable urban planning, in an international perspective.

Author Response

I thank the reviewer for his remark.  I completely agree that the relation to climate change and sustainable urban planning is important to be mentioned. This was added now in the introduction and conclusion.

Reviewer 2 Report

I would recommend that you reduce the references, as they are a third of the article, and expand the comments on each of them.

Author Response

Sorry about that. The review was expanded to cover the interest of the different publications citated.

Reviewer 3 Report

The review paper addresses pollution accumulation and health outcomes in energy efficient buildings with and without mechanical ventilation. This is an important and very current issue. However, it is not felt the review paper adequetly addresses this aim, either through providing enough evidence or communicating the results of other studies in a clear and balanced way. 

A number of studies are cited, but these are small samples in a complex field. A more robust and wider reaching literature search is required. Alongside this a clearer structure should be sought, providing a clear balance f the evidence in mechanical and non-mechanical buildings. Where clear conclusions cannot be drawn, or where evidence is conflicting, these gaps should be highlighted. The conclusions at present demonstrate there is little strong outcomes from this review in its current form.

Air quality guidance levels are not suitably referenced or discussed. There are different guidelines set by different bodies across the world. 

Author Response

We completely agree that there is a complexe field. The aim was precised and the conclusions were improved to target what this review is bringing in the field

Author Response

We thank the reviewer for the remarks .

The annotations were modified and the units justified

Round 2

Reviewer 4 Report

Many pollutants are removed from original manuscript. O3, SO2, CO and Naphthalene are removed from the study, as a result the manuscript becomes less informative. 6th column of Table 1 is removed which is not logical.